# POI-based Traffic Generation via Supervised Contrastive Learning on Reconstructed Graph

## Abstract

Traffic flow generation problem under realistic scenarios has raised more and more attention in recent years. This problem aims at generating traffic flow without using historical traffic data. Since road network and POI data can provide a more comprehensive picture of traffic patterns, most previous methods use both or either of them to generate traffic flow. However, roadnet graph in real-world has bias and abnormal structure, which will influence the performance of traffic generation. Previous traffic generation models directly receive real-world roadnet graph with map-match POI data as input and then use an end-to-end loss for training, which could not model the complex relationship between POI and traffic in a proper way. Different from prior methods, we propose a novel POI-based **T**raffic **G**eneration model via **S**upervised **C**ontrastive learning on **R**econstructed graph, termed as **TG-SCR**, which combines POI data and road network data to generate the distribution of traffic flows. Our model has two novel modules: a graph reconstruction module and a POI supervised contrastive module. The structural module includes a k-NN graph builder and a k-NN graph aggregator, which is used to reconstruct the original roadnet graph into a k-NN graph and reform POI feature. The contrastive module is used to model the relationship between POI feature and traffic flow. Extensive experiments show that our model outperforms other baseline methods on four real-world datasets.

## 1 Introduction

Recently, traffic generation problem has become an increasingly popular research topic due to the growing importance of intelligent transportation systems and the availability of large amounts of traffic data. It has important implications for traffic management, safety, and environmental sustainability. Generating traffic flow precisely can help transportation authorities and planners make accurate decisions about traffic flow management and infrastructure investment. For example, predicting traffic volume and congestion can help officials adjust traffic signal timings, optimize route planning, and manage demand for public transportation.

For a long time, people use time series forecasting methods to predict traffic flows, such as traditional statistical methods like ARIMAWilliams & Hoel (2003). In recent years, deep learning models such as RNN-based modelsVan Lint et al. (2002), LSTM-based modelsZhao et al. (2017); Fu et al. (2016); Chen et al. (2016) and STGNN-based modelsYu et al. (2017a); Li et al. (2017) have also been applied to traffic prediction tasks with promising results. These models can capture spatial dependencies between traffic variables and learn complex traffic patterns in high-dimensional traffic data.

In this paper, we focus on "Traffic Generation Task", which is completely different from "Traffic Prediction Task". The biggest difference between these two tasks is that traffic prediction models predict the future traffic based on historic traffic data, while traffic generation models generate the future traffic based on other non-traffic data such as POI data or check-in data. For instance, if we want to forecast the traffic flow in the next week, traffic prediction models need historical traffic data or any data in the same format as the volume to be predicted. However, traffic generation models simply require static data to simulate the real environment and then generate traffic flow. Compared

to traffic prediction task, traffic generation task is more meaningful in real-world for the following reasons. Firstly, normal traffic prediction models require large scale of historic traffic data to improve their predicting performance. However, it is not easy to collect real-world traffic, especially in small cities. Secondly, the performance of historical traffic-based traffic prediction models become worse if the environment changes, such as road construction, new shopping malls and so on. Instead, traffic generation task is to explore the deep principles of how traffic flow generates, for which these situations can be handled. Therefore, it is meaningful to study on traffic generation task.

Despite the rapid development of deep learning models in traffic generation domain, most of previous traffic generation models face two challenges. The first challenge is the abnormal structures in real-world roadnet graph. Since road networks provide information on the physical layout of roads, such as their length and connectivity, most previous traffic generation models directly use real-world roadnet graph as backbone and map spatio-temporal data on it, so as to explore how traffic flow is likely to be distributed across different routes and intersections. However, roadnet graphs in real-world do not always hold the basic law that "neighbor nodes have similar features". Previous GNN-based models could not handle this issue, resulting in accumulation of bias in the training procedure, therefore reducing the generating quality.

The second challenge is how to train the traffic generation model in a proper way. Most previous traffic generation models focus on imposing constraints on the final predictions in an end-to-end style, such as using Cross-Entropy loss or MSE loss, but do not explicitly consider the representations learned by the model. More specifically, the weights of neural networks with end-to-end loss are optimized simply based on the output errorm which means the network is only capable of capturing the output error rather than capturing the complex relationships between POI features. This will lead to overfitting and reduced accuracy when dealing with more complex data. Thus, previous models do not make good use of the soft label information of traffic data.

In this paper, we propose a new deep learning model, which combines POI data and road network data to predict the distribution of traffic flows, thus successfully solving the problem of generating traffic flows in areas without historical traffic data. Extensive experiments show that our designed model achieves the best performance on four real-world datasets.

Our main contributions can be summarized as below:

- **Structural Module.** We add a structural module including a k-NN Graph Builder and a k-NN Graph Aggregator to reform the original roadnet graph into a k-NN Graph and revise the original POI feature. By doing so, model successfully eliminate bias and anomalies in the real-world roadnet graph.

- **Contrastive Module.** In this module, we add a supervised contrastive loss to learn a regression-aware representation by contrasting POI feature embedding against other nodes in a batch based on their target distance. By doing so, our model explicitly hold the former similarity relationships between samples to optimize the representation for the traffic generation task.

- **Evaluation.** Under extensive experiments on four real-world datasets, we show that TG-SCR provides consistent boosts in traffic generation performance.

## 2    RELATED WORKS

### 2.1    TRADITIONAL NON-DEEP-LEARNING METHODS

Time series models have a wide range of applications in the domain of traffic prediction. Conventional time series models such as ARIMAAhmed & Cook (1979) are first used for traffic flow prediction. After that, extended variants such as SARIMIAWilliams & Hoel (2003), KARIMAVan Der Voort et al. (1996), STARIMASun et al. (2005); Kamarianakis & Prastacos (2003) and ARIMAXWilliams (2001) are also used for predicting traffic. Besides, researchers also use other parametric model like Kalman filterGuo et al. (2014) and non-parametric model like k-nearest neighbors modelsCai et al. (2016), bayesian networkSun et al. (2005), SVRWu et al. (2004) to predict traffic flow. There are also traffic prediction methods that are based on transportation laws. Zhou et al. asu-trans-ai lab (2020) propose Grid2Demand to generate traffic flow in gridded areas based on

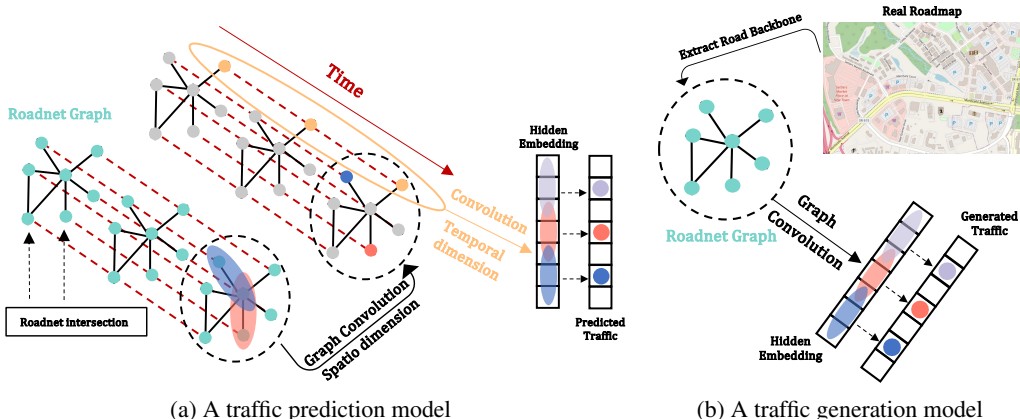

(a) A traffic prediction model      (b) A traffic generation model

Figure 1: Difference between conventional traffic prediction problem and traffic generation problem studied in this paper.

the gravity model. However, performance of these methods can be limited by their assumptions and biases, and traditional non-deep-learning methods may struggle with high-dimensional data or complex relationships between variables.

## 2.2 DEEP LEARNING MODELS WITHOUT GRAPH

With the development of deep neural networks, many researchers try to apply different deep learning methods to solve the traffic generation problem. DBN-SVRLi & Wang (2017) uses multiple RBM models to extract features from the data and a support vector regression classifier is connected on top to predict traffic flow. Researchers first use RNN-based models such as LSTMZhao et al. (2017); Fu et al. (2016), R-LSTMJia et al. (2017) and LSTM-GRUFu et al. (2016) to capture the time dependence of traffic flow. In order to consider spatial correlations in traffic flow, researchers also start to use CNN-based models such as UrbanFlowZhang et al. (2016), DeepCNNMa et al. (2017), ST-ResNetZhang et al. (2017). Afterwards, hybrid models combining CNN and RNN are widely used to solve the traffic prediction task, such as DNN-BTFWu et al. (2018), HMDLFDu et al. (2018), Deep-LSTMYu et al. (2017c) and SRCNYu et al. (2017b). However, these deep learning models have poor generalization performance and are difficult to use to predict traffic flow under real-world datasets.

## 2.3 GRAPH NEURAL NETWORK MODELS

In recent years, there has been an increasing enthusiasm in the domain of deep learning for Graph Neural Networks (GNNs). GNNs have emerged as a new and powerful approach for handling graph-structured data, which make it useful in solving traffic prediction task. GNN-based models such as DCRNNLi et al. (2017), STGCNYu et al. (2017a), STFGNNLi & Zhu (2021), TrajNetHui et al. (2021), STDN and MVGCNFu et al. (2022). Furthermore, in order to capture the dynamic relationship between traffic flow and other influence factors, more models such as AGCRNBai et al. (2020), DGCRNLi et al. (2023), DMSTGCNHan et al. (2021) and DSTAGNNLan et al. (2022) are proposed. However, under real-world dataset, the roadnet graph which GNN-based models research on may have bias, resulting in inaccuracy of traffic flow prediction.

## 3 PRELIMINARY

We start with formally introducing the problem of traffic generation. A city-level road network graph $\mathbf{G} = (\mathbf{V}, \mathbf{E})$ is given, where $n_i$ denotes the $i$-th node of the graph and $e_j$ denotes the $j$-th edge. $\mathbf{G}$ carries a node-level POI features namely $\mathbf{X}$. For each node in $\mathbf{G}$, $x_i$ represents a 24-dimensional Point of Interest (POI) feature vector, where each dimension corresponds to the number of one category of POI points within a 1km radius around $n_i$. Similarly, $y_i$ represents a 24-dimensional

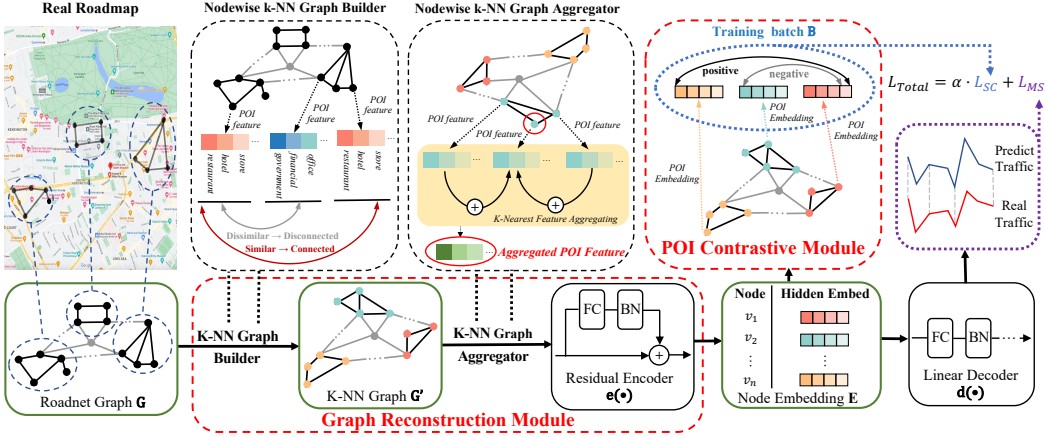

Figure 2: The overall architecture of TG-SCR.

traffic vector, where each dimension corresponds to the number of arrival points within a 1km radius around $n_i$ for an one-hour time slice.

Different from previous traffic prediction task based on spatio-temporal data, the traffic generation task studied in this paper simply focuses on the spatio dimension. Figure 1a shows how a common spatio-temporal model predicts traffic flow. In the conventional traffic prediction task, one main way to achieve the goal is to combine graph convolution with recurrent neural network to capture both spatial and temporal dependencies. Figure 1b shows a common model to solve the traffic generation task, where only graph convolution is used to generate traffic flow.

The traffic generation task can be defined as below:

**Problem Definition 1 (Traffic Generation)** *Given a roadnet graph* **G** *along with its POI feature* **X** *that has been map-match on* **G***, the objective of traffic generation task is to generate an one-day traffic flow* **Y***. Suppose* $f(\cdot)$ *is a model that maps POI feature to traffic, this traffic generation task can be formulated as:*

$$Y = f(\boldsymbol{G}, \boldsymbol{X}) \tag{1}$$

## 4 METHODOLOGY

### 4.1 TRAFFIC GENERATION

To generate traffic flow based on roadnet graph along with POI data, we propose a novel POI-based traffic generation model via supervised contrastive learning on reconstructed graph, called **TG-SCR**. TG-SCR has two modules: a graph reconstruction module and a POI contrastive module, focusing on structural view and POI contrastive view respectively. The first graph reconstruction module has two components: a k-NN Graph Builder and a k-NN Graph Aggregator. k-NN graph builder is used for converting the original roadnet graph into a k-NN graph, while the k-NN Graph Aggregator is used for aggregating POI feature to generate reforming feature. In the second POI supervised contrastive module, motivated from the insight that "Similar POI patterns generate similar traffic flow", we propose a supervised contrastive loss with traffic feature as soft label and then use it to train the residual encoder. By incorporating these two modules, our model can better distinguish between different POI features and their corresponding traffic patterns. The overall architecture of TG-SCR is shown in Figure 2.

### 4.2 GRAPH RECONSTRUCTION MODULE

Under realistic scenarios, roadnet graphs do not conform to the laws of general graph datasets. As is shown in Figure 3, it can be found that one of the laws that should be satisfied in the regular

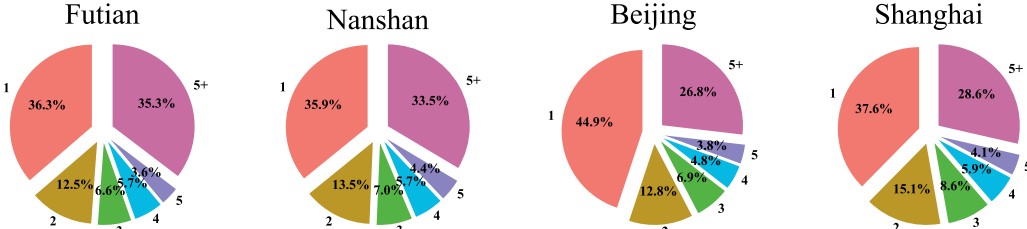

Figure 3: The pie chart describing hop number ratio of shortest path of node pairs with nearest POI feature in Futian, Nanshan, Beijing and Shanghai.

datasets, i.e., "the closer the points are, the closer the relationship is", is not fully satisfied in real-world roadnet graph. It can be seen from Figure 3 that about forty to fifty percent of the node pairs with nearest POI feature are not connected in the roadnet graph (the shortest path is greater than one hop), which contradicts the human intuition that connected node pairs should have similar POI feature. Therefore, corrections need to be made to the original roadnet graph when using it for traffic prediction. Graph Reconstrcuction module includes a k-NN Graph Builder and a k-NN Graph Aggregator. The k-NN graph builder converts the original roadnet graph $G$ into a k-nearest neighbor graph $G'$. Assume the $i$-th node is denoted as $n_i$ with its POI feature $X_i$ and traffic flow $Y_i$. Each dimension of $X_i$ or $Y_i$ is denoted as $x_i^d$ or $y_i^d$ ($d \leq 24$). For the k-NN graph building step, we first compute the L2 distance $l$ between the anchor node and other nodes (suppose the $j$-th node):

$$l_{i,j} = ||x_i - x_j|| = \sqrt{\sum_d \left(x_i^d - x_j^d\right)^2} \tag{2}$$

Suppose the L2 distances between $n_i$ and other nodes constitutes a set $\mathbb{L}_i$. We select $k$ nodes with minimum L2 distance and connect them pairwisely in the newly formed k-NN graph. These k nodes will be reformed into the neighbor nodes of $n_i$ in k-NN graph. The set of these nodes is denoted as $\mathbb{K}_i$. This step can be formulated as:

$$\mathbb{K}_i = \{n_j \in N : |\mathbb{L}_i \cap (-\infty, l_{i,j})| < k\} \tag{3}$$

After that, a k-NN Graph Aggregator is applied to transform the raw POI feature $X$ into aggregated POI feature $X'$ by adding the nearest POI feature in $G'$. The aggregating operation can be formulated as below:

$$X'_i = X_i + \sum_{j \in \mathbb{K}_i} X_j \tag{4}$$

### 4.3 POI CONTRASTIVE MODULE

The relationship among POI, roadnet and traffic is complicated. Most previous models simply use an end-to-end loss to optimize the mapping relations. However, the correlation of POI embeddings between different nodes belonging to the same POI pattern is not taken into account in end-to-end loss function. Therefore, we consider improving the expressiveness using the soft label information of traffic data. Since the insight of contrastive loss is to make the hidden embeddings of samples in the same class close to each other, we propose a supervised contrastive loss that use traffic data as soft label, which can ensure distances in the embedding space are ordered according to distances in the label space.

Suppose in a training batch $B$ with $N$ nodes, the randomly selected anchor node in $B$ is denoted as $n_i$, its traffic and POI embedding are denoted as $y_i$ and $v_i$ respectively. The predicted traffic of this node is denoted as $\hat{y}_i$. We first select $c$ nearest nodes in $B$ based on their traffic distance and treat them as positive set $\mathbb{P}$, then the rest nodes in $B$ are classified to negative set $\mathbb{N}$. The procedure of generating positive set and negative set can be formulated as below:

$$\mathbb{P} = \{n_j \in B : |Y \cap (-\infty, ||y_i - y_j||)| < c\} \tag{5}$$
$$\mathbb{N} = \{n_k\}, k \neq i, k \in B, k \notin \mathbb{P} \tag{6}$$

Based on the positive set defined in Eq (5) and the negative set defined in Eq (6), our POI supervised contrastive loss $L_{SC}$ can be formulated as:

$$\mathcal{L}_{\text{SC}} = -\frac{1}{N} \log \frac{\sum\limits_{j \in \mathbb{P}} \exp\left(\text{sim}\left(\boldsymbol{v}_i, \boldsymbol{v}_j\right) / \tau\right)}{\sum\limits_{k \in \mathbb{N}} \exp\left(\text{sim}\left(\boldsymbol{v}_i, \boldsymbol{v}_k\right) / \tau\right)} \tag{7}$$

where $\tau$ represents the temperature coefficient. In $L_{SC}$ loss, any two nodes can be thought of as a positive pair or a negative pair, depending on their similarities in traffic flow. We define positive and negative samples in a relative way, based on the distance between their traffic and that of anchor node.

## 4.4 Loss Summary

In addition to the contrastive loss mentioned above, we define another loss to describe the error between predicted and real traffic flow from a global perspective. This loss is used to measure the mean square error between predicted traffic $\hat{Y}$ and real traffic $Y$. It can be formulated as below:

$$L_{MSE} = MSELoss(\hat{Y}, Y) = \frac{1}{N} \sum_{i=1}^{N} (\hat{y}_i - y_i)^2 \tag{8}$$

The final loss combines the $L_{SC}$ term and $L_{MSE}$ term:

$$L_{Total} = \alpha \cdot L_{SC} + L_{MSE} \tag{9}$$

where $\alpha$ is a hyperparameter.

To alleviate the gradient vanishing problem during training procedure, we simply train the $L_{MSE}$ for the first five hundred epochs, and then add $L_{SC}$ into training. The training procedure of TG-SCR can be expressed as below:

---

**Algorithm 1** Training procedure of TG-SCR

---

**Input:** Roadnet graph $G$ with map-matched POI feature $X$, encoder $e(\cdot)$ and decoder $d(\cdot)$ parameterized with $\theta$
1: Build k-NN graph $G'$ from $G$ based on Eq (3)
2: Get Aggregated POI feature $X'$ from $X$ based on Eq (4)
3: Compute POI embeddings: $E' = e(X')$
4: **repeat**
5:     Sample a training batch $B$ including $N$ nodes
6:     Define positive set $\mathbb{P}$ and negative set $\mathbb{N}$ based on Eq (5) and Eq (6)
7:     Compute $L_{Total}$ based on Eq (9)
8:     Back Propagation
9:     Update $\theta$ based on Eq (7)
10: **until** $L_{Total} < \epsilon$
**Output:** Predicted traffic $\hat{Y}$

---

## 5 Experiment

### 5.1 Experiment Setup

#### 5.1.1 Datasets

We evaluate our model on four real-world datasets of Futian, Nanshan, Shanghai and Beijing. In these four datasets, nodes and edges in the graph are roadnet junctions and roads extracted from OpenStreetMap. Node features consist of POI data, which are queried from Baidu Map with API tools. The traffic data of Futian, Nanshan, Beijing and Shanghai is downloaded from Baidu (2019c), Baidu (2019a) and Baidu (2019b). For POI data, according to Baidu, we divide it into twenty-four categories. For traffic data, we split the one-day trajectory data using our-hour as grain size. Since traffic data and POI data are sparse in cities, we filter nodes with excessively sparse traffic or POI data to reduce the sparsity of graphs. Details of these four datasets are shown in Table 1.

| Indicator | Futian | Nanshan | Beijing | Shanghai |
|---|---|---|---|---|
| # Nodes | 42,018 | 20,969 | 86,011 | 94,661 |
| # Edges | 52,621 | 26,929 | 115,728 | 125,290 |
| Mean POI Count | 177.97 | 165.22 | 41.99 | 74.70 |
| Standard Deviation of POI Count | 264.63 | 239.39 | 37.79 | 137.51 |
| Mean Traffic Volume | 420.35 | 231.17 | 40.12 | 30.17 |
| Standard Deviation of Traffic Volume | 337.77 | 155.00 | 31.94 | 24.52 |

Table 1: Basic information of four real-world datasets.

### 5.1.2 BASELINE METHODS

We compare our model with three categories of methods. First category includes several traditional machine learning methods used to solve regression problems, such as DecisionTree (DTree)Quinlan (1986), RandomForest (RF)Breiman (2001), XGBoostChen & Guestrin (2016) and SVRCortes & Vapnik (1995).The second category is methods that based on Graph Neural Networks, such as GCNKipf & Welling (2016), GraphSAGEHamilton et al. (2017), GATVelickovic et al. (2017) and GINXu et al. (2018). The third category includes other models that are used for generating traffic flow based on POI data, such as DeepFlowGen (DFG)Shao et al. (2021) and DeepCrowdJiang et al. (2021). The results of the comparison between our model and baselines are shown in Table 2.

### 5.1.3 EXPERIMENT SETTINGS

We use three common metrics to evaluate the prediction error between predict traffic and real traffic, which are Root Mean Square Error (RMSE), Mean Absolute Error (MAE) and Mean Absolute Percentage Error (MAPE). Since MAPE cannot be calculated when the true value equals zero, we revise the MAPE metric by masking locations where the real traffic flow is zero in this timestep then evaluating the rest. To overcome the issue of gradient vanishing, we add $L_{SC}$ into training after five hundred epochs. The parameters of all methods have been tuned through a grid search. Code is available at https://anonymous.4open.science/r/TG-SCR-D0BE/.

### 5.2 EXPERIMENT RESULTS

The results of the comparison between our model and baselines are shown in Table 2. On all of these four real-world datasets with different scales, TG-SCR outperforms other baseline methods in all three evaluation metrics: RMSE, MAE and MAPE. The experiment results show that for all the real-world datasets with different data scale, our model achieve the best performance. Notably, Table 2 shows that some GNN-based methods like GCN performs badly in generating traffic. This is mainly due to the fact that these methods are influenced by abnormal structures in the real-world roadnet graph. GNN's message passing mechanism leads to errors where the predicted traffic of some specific area (e.g. hospital, school) are severely influenced by some more distant areas. TG-SCR overcome this problem by reconstructing the original roadnet graph and thus avoiding the flaws brought by message passing.

### 5.3 ABLATION STUDY

As discussed in Section 4.1, TG-SCR includes two novel modules: a graph reconstruction module (denoted as GRM) and a POI supervised contrastive module (denoted as SCM). We perform ablation study to examine the effect of these two modules in TG-SCR. We consider two variants of TG-SCR: **(1)** TG-SCR without graph reconstruction module (abbreviated as TR-SCR w/o GRM). **(2)** TG-SCR without graph reconstruction module and POI supervised contrastive module (abbreviated as TG-SCR w/o GRM & SCM). Experiment results on Futian dataset are shown in Table 3, from which we could observe that TG-SCR has performance gain over two variants, proving the significance of the proposed graph reconstruction module and POI supervised contrastive module.
The improvement induced by adding the POI supervised contrastive module (SCM) is due to that adding a supervised contrastive loss term helps model to learn to complicated relationship between

| Method | Futian | | | Nanshan | | |
|---|---|---|---|---|---|---|
| | RMSE | MAE | MAPE | RMSE | MAE | MAPE |
| RF | 64.28 | 52.83 | 35.13% | 74.72 | 30.66 | 24.88% |
| XGBoost | 76.13 | 62.01 | 36.37% | 67.13 | 36.81 | 26.76% |
| SVR | 262.91 | 157.56 | 64.06% | 100.18 | 53.59 | 51.84% |
| GCN | $173.57 \pm 4.17$ | $102.46 \pm 2.61$ | $49.76\% \pm 3.54\%$ | $92.57 \pm 0.93$ | $66.50 \pm 0.43$ | $25.38\% \pm 1.39\%$ |
| GraphSAGE | $102.64 \pm 3.04$ | $74.62 \pm 2.56$ | $42.94\% \pm 2.59\%$ | $90.18 \pm 1.00$ | $63.59 \pm 0.46$ | $24.73\% \pm 0.91\%$ |
| GAT | $92.47 \pm 1.57$ | $68.47 \pm 0.47$ | $39.83\% \pm 0.82\%$ | $71.48 \pm 1.31$ | $31.09 \pm 0.55$ | $21.41\% \pm 1.57\%$ |
| GIN | $79.33 \pm 1.11$ | $58.92 \pm 0.54$ | $40.97\% \pm 0.60\%$ | $65.25 \pm 0.62$ | $29.59 \pm 0.28$ | $21.08\% \pm 0.79\%$ |
| DFG | $82.39 \pm 0.46$ | $63.64 \pm 0.37$ | $38.61\% \pm 0.65\%$ | $72.46 \pm 0.57$ | $36.47 \pm 0.30$ | $23.35\% \pm 0.59\%$ |
| DeepCrowd | $92.46 \pm 1.38$ | $67.54 \pm 0.70$ | $38.45\% \pm 0.86\%$ | $74.36 \pm 0.37$ | $36.46 \pm 0.23$ | $23.72\% \pm 0.34\%$ |
| **TG-SCR** | $\mathbf{53.19 \pm 0.40}$ | $\mathbf{41.03 \pm 0.21}$ | $\mathbf{30.06\% \pm 0.69\%}$ | $\mathbf{51.56 \pm 0.42}$ | $\mathbf{27.43 \pm 0.24}$ | $\mathbf{19.05\% \pm 0.29\%}$ |

| Method | Beijing | | | Shanghai | | |
|---|---|---|---|---|---|---|
| | RMSE | MAE | MAPE | RMSE | MAE | MAPE |
| RF | 9.03 | 4.56 | 17.62% | 6.62 | 3.02 | 13.04% |
| XGBoost | 14.52 | 7.33 | 29.24% | 7.81 | 4.50 | 18.89% |
| SVR | 19.12 | 10.16 | 36.34% | 25.05 | 13.57 | 46.84% |
| GCN | $20.46 \pm 1.46$ | $13.49 \pm 0.84$ | $56.20\% \pm 1.95\%$ | $18.34 \pm 0.87$ | $11.52 \pm 0.33$ | $42.71\% \pm 3.73\%$ |
| GraphSAGE | $22.56 \pm 1.36$ | $14.22 \pm 0.83$ | $53.37\% \pm 1.86\%$ | $14.36 \pm 1.31$ | $9.64 \pm 0.83$ | $41.69\% \pm 5.31\%$ |
| GAT | $18.92 \pm 1.30$ | $12.38 \pm 0.56$ | $33.27\% \pm 2.02\%$ | $9.44 \pm 1.03$ | $6.82 \pm 0.95$ | $28.59\% \pm 3.51\%$ |
| GIN | $10.29 \pm 0.99$ | $6.22 \pm 0.40$ | $24.68\% \pm 2.42\%$ | $6.57 \pm 0.17$ | $3.79 \pm 0.06$ | $21.06\% \pm 1.42\%$ |
| DFG | $3.48 \pm 0.16$ | $2.29 \pm 0.06$ | $9.55\% \pm 0.21\%$ | $2.39 \pm 0.04$ | $1.74 \pm 0.02$ | $9.54\% \pm 0.60\%$ |
| DeepCrowd | $10.29 \pm 1.04$ | $6.22 \pm 0.46$ | $24.86\% \pm 2.89\%$ | $6.57 \pm 0.09$ | $4.25 \pm 0.03$ | $16.63\% \pm 0.92\%$ |
| **TG-SCR** | $\mathbf{1.47 \pm 0.07}$ | $\mathbf{1.10 \pm 0.05}$ | $\mathbf{5.12\% \pm 0.14\%}$ | $\mathbf{1.67 \pm 0.03}$ | $\mathbf{1.26 \pm 0.02}$ | $\mathbf{7.31\% \pm 0.83\%}$ |

Table 2: Performance comparison between TG-SCR and other baseline methods on four real-world datasets. Results shows that our model outperforms other baseline methods in all four real-world datasets. All experiments are repeated for five times, and the mean and standard deviation ($\pm$) are reported.

| Method | Futian | | | Nanshan | | |
|---|---|---|---|---|---|---|
| | RMSE | MAE | MAPE | RMSE | MAE | MAPE |
| **TG-SCR** | $\mathbf{53.19 \pm 0.40}$ | $\mathbf{41.03 \pm 0.21}$ | $\mathbf{30.06\% \pm 0.69\%}$ | $\mathbf{51.56 \pm 0.42}$ | $27.43 \pm 0.24$ | $\mathbf{19.05\% \pm 0.29\%}$ |
| TG-SCR w/o GRM | $55.67 \pm 0.42$ | $42.63 \pm 0.21$ | $31.28\% \pm 1.66\%$ | $52.72 \pm 0.61$ | $\mathbf{27.19 \pm 0.23}$ | $19.28\% \pm 1.08\%$ |
| TG-SCR w/o GRM & SCM | $92.46 \pm 1.38$ | $67.54 \pm 0.70$ | $38.40\% \pm 0.86\%$ | $72.46 \pm 0.57$ | $36.47 \pm 0.42$ | $23.58\% \pm 8.36\%$ |

Table 3: Performance comparison between TG-SCR and its variants. All experiments are repeated for five times, and the mean and standard deviation ($\pm$) are reported.

POI features and traffic. The performance boosted by adding graph reconstruction module (GRM) is due to reconstructing graph saves model from being influenced by anomalous structures in the original real-world roadnet graph.

### 5.4 PARAMETER SENSITIVE ANALYSIS

There are two important hyperparameters that should be conducted, which are the combination coefficient $\alpha$ named in Eq (9) and the neighbor aggregator number $k$. We test the sensitivity of $\alpha$ by varying it from 0.001 to 0.1 using the regular experiment setting. Also, the sensitivity of $k$ are tested by varying it from 1 to 10. The corresponding RMSE results are respectively shown in Figure 4a with blue lines. The blue shaded area in each line chart indicates the error range of the corresponding standard deviation. Among eight line chart in Figure 4a, $\alpha$ is senstive merely in Futian dataset, and $k$ is not sensitive in all the four dataset. We find that the performance of TG-SCR is generally stable with the change of $\alpha$ and $k$, which means the performance of TG-SCR is not sensitive to $\alpha$ or $k$.

### 5.5 CASE STUDY

In case study, we select part of data in Shanghai dataset and visualize the POI embedding of GCN and TG-SCR using t-SNE dimensionality reduction algorithm. The visualization results are shown in Figure 4b. The two locations circled on the map with dashed boxes are far in distance but similar in POI distribution. It can be seen that the POI embeddings of nodes in these two locations more similar than that of GCN, which shows that our model can capture feature similarities and filter out abnormal edges in the original graph.

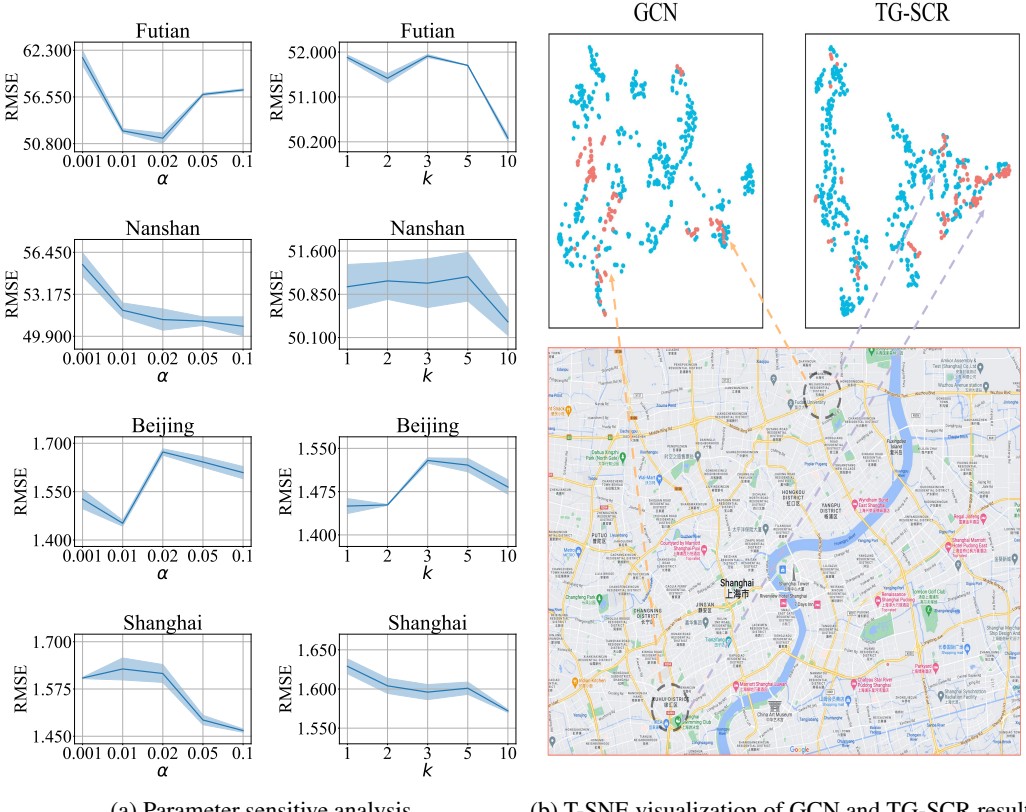

(a) Parameter sensitive analysis      (b) T-SNE visualization of GCN and TG-SCR results

Figure 4: **(a).** Two parameter sensitivity analysis on four real-world datasets. The corresponding RMSE results are respectively shown using line charts. The blue shaded area in each line chart indicates the error range of the corresponding standard deviation. **(b).** T-SNE visualization of POI embeddings of GCN and TG-SCR on part of Shanghai dataset.

## 6 CONCLUSION

In this paper, we have researched on the traffic generation problem without using historical traffic data. We demonstrate that the original roadnet graph has abnormal structures and it is not enough to learn the complex relationship between POI data and traffic flow with simply end-to-end loss. To address these issues, we propose TG-SCR, a novel POI-based traffic generation model via supervised contrastive learning on reconstructed graph. Our model includes a graph reconstruction module and a POI supervised contrastive module. First, we propose a graph reconstruction module to reconstruct the original roadnet graph into a k-NN graph. After that, a POI supervised contrastive module is conducted to model the relationship between POI feature and traffic flow. Moreover, we conduct experiments on four real-world datasets. Extensive experiments indicate that our model outperforms other baseline methods. In conclusion, by combing k-NN graph reconstruction mechanism and supervised contrastive loss, TG-SCR can generate traffic flow with high quality based on POI data in real-world datasets.

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

# A APPENDIX

## A.1 EXPERIMENTS ON SPATIO-TEMPORAL MODELS

To further illustrate the power of TG-SCR in solving the traffic generation problem, we additionally conduct experiments on several spatio-temporal models as well as their variants. Taking DCRNN as an example, $DCRNN_{mlp}$ denotes that the GRU and RNN structures in the model are replaced with fully connected layers. $DCRNN_{transformer}$ adds a self-attention mechanism and positional encoding based on $DCRNN_{mlp}$. The results of the comparison between our model and other spatio-

| Method | Futian | | | Nanshan | | |
|---|---|---|---|---|---|---|
| | RMSE | MAE | MAPE | RMSE | MAE | MAPE |
| DCRNN | 442.5912 | 50.3046 | 43.04% | 380.4306 | 45.0945 | 40.33% |
| $DCRNN_{mlp}$ | 133.1649 | 38.0537 | 24.42% | 124.486 | 32.2951 | 24.82% |
| $DCRNN_{transformer}$ | 368.4053 | 45.5119 | 40.48% | 329.595 | 40.0576 | 34.73% |
| STGCN | 285.411 | 43.6264 | 36.81% | 274.522 | 35.381 | 30.46% |
| $STGCN_{mlp}$ | 109.4752 | 34.4618 | 21.03% | 88.4164 | 30.0691 | 21.9% |
| $STGCN_{transformer}$ | 405.0078 | 48.5714 | 41.92% | 468.2043 | 49.1706 | 42.98% |
| D2STGNN | 481.4794 | 53.5861 | 44.55% | 522.9714 | 54.633 | 45.73% |
| $D2STGNN_{mlp}$ | 172.5711 | 40.3273 | 32.08% | 157.582 | 33.6911 | 27.4% |
| $D2STGNN_{transformer}$ | 406.8035 | 47.0052 | 42.79% | 470.4691 | 49.698 | 42.23% |
| **TG-SCR** | **51.5639** | **27.4304** | **19.57%** | **53.275** | **27.947** | **18.45%** |

| Method | Beijing | | | Shanghai | | |
|---|---|---|---|---|---|---|
| | RMSE | MAE | MAPE | RMSE | MAE | MAPE |
| DCRNN | 1003.7166 | 530.1937 | 1019.42% | 1421.4306 | 768.0945 | 1359.5% |
| $DCRNN_{mlp}$ | 22.4629 | 8.4756 | 26.58% | 13.486 | 4.2951 | 47.82% |
| $DCRNN_{transformer}$ | 842.5207 | 481.3813 | 857.3% | 793.5288 | 558.5023 | 750.19% |
| STGCN | 693.5531 | 423.1444 | 729.45% | 1146.5004 | 603.2289 | 1145.73% |
| $STGCN_{mlp}$ | 14.5814 | 6.5035 | 13.76% | 9.0463 | 3.5702 | 38.85% |
| $STGCN_{transformer}$ | 515.3719 | 274.364 | 735.4% | 689.421 | 520.471 | 663.52% |
| D2STGNN | 1280.0422 | 581.451 | 1257.26% | No Convergence | No Convergence | No Convergence |
| $D2STGNN_{mlp}$ | 38.5926 | 14.5609 | 103.91% | 45.487 | 26.9913 | 351.46% |
| $D2STGNN_{transformer}$ | No Convergence | No Convergence | No Convergence | No Convergence | No Convergence | No Convergence |
| **TG-SCR** | **1.5364** | **1.1647** | **4.28%** | **1.6713** | **1.265** | **19.05%** |

Table 4: Performance comparison between TG-SCR and other spatio-temporal models with their variants on four real-world datasets. Results shows that TG-SCR outperforms other spatio-temporal models and their variants in all four real-world datasets.

temporal models are shown in Table 4. On all of these four real-world datasets with different scales, TG-SCR outperforms other spatio-temporal models as well as their variants. It can be seen that even when compared to spatio-temporal convolutional models, TG-SCR is powerful at generating traffic flows.

## A.2 COMPARISON WITH TRAFFIC FLOW DATASETS

Compared with the traditional spatio-temporal traffic dataset, these four real-world datasets in this paper is much more larger in scale and therefore more consistent with real-life traffic prediction scenarios. The basic information of the four real-world datasets and some common spatio-temporal traffic datasets is shown in Table 5.

| Indicators | Futian | Nanshan | Beijing | Shanghai | METR-LA | PEMS-03 | PEMS-04 | PEMS-07 | PEMS-08 | PEMS-BAY |
|---|---|---|---|---|---|---|---|---|---|---|
| # Nodes | 42,018 | 20,969 | 86,011 | 94,661 | 325 | 358 | 307 | 883 | 170 | 207 |
| # Edges | 52,621 | 26,929 | 115,728 | 125,290 | 2369 | 2268 | 2591 | 2704 | 2914 | 1515 |

Table 5: Basic information of four real-world datasets and other traffic flow datasets.

## A.3 EXPERIMENTS ON EFFICIENCY

We also experimentally analyzed the efficiency of TG-SCR. Table 6 shows the training time of TG-SCR and other spatio-temporal models on four real-world datasets. The training time of TG-SCR is significantly shorter than that of the other spatio-temporal graph models and competitable with

that of their MLP variants. These results fully demonstrate that TG-SCR not only achieves good performance, but also has higher efficiency compared to other spatio-temporal models.

| Model | Futian | Nanshan | Beijing | Shanghai |
|---|---|---|---|---|
| DCRNN | 5130 | 5715 | 4812 | 4606 |
| $\text{DCRNN}_{mlp}$ | 1836 | 1900 | 1105 | 1257 |
| $\text{DCRNN}_{transformer}$ | 12645 | 10504 | 9350 | 17559 |
| STGCN | 2691 | 2588 | 2042 | 2539 |
| $\text{STGCN}_{mlp}$ | 473 | 492 | 500 | 588 |
| $\text{STGCN}_{transformer}$ | 10572 | 11469 | 15058 | 15770 |
| D2STGNN | 4615 | 5028 | 3711 | - |
| $\text{D2STGNN}_{mlp}$ | 542 | 761 | 909 | 827 |
| $\text{D2STGNN}_{transformer}$ | 15294 | 13460 | - | - |
| **TG-SCR** | 628 | 733 | 606 | 675 |

Table 6: Training time (second) of TG-SCR and other spatio-temporal models with their variants on four real-world datasets.

