# OpenReview forum: "POI-based Traffic Generation via Supervised Contrastive Learning on Reconstructed Graph"
_ICLR.cc/2024/Conference — ICLR 2024 Conference Withdrawn Submission_

### Official Review · Reviewer_rnVi · 2023-10-28

**Soundness:** 3 good
**Presentation:** 3 good
**Contribution:** 2 fair
**Rating:** 3
**Confidence:** 4

**Summary:**

This paper proposes a model called TG-SCR which borrows the POI information and the road network to build a reliable graph to realize traffic generation. Extensive experiments are conducted.

**Strengths:**

1. The paper is well-presented and well-organized.
2. The paper proposed a novel graph based method aiming to perform traffic generation without using any prior data.
3. Extensive experiments are conducted.

**Weaknesses:**

1. This paper try to generate traffic without using prior traffic data, not sure why "traffic generation" is better than "traffic prediction". Intuitively, using prior data to do prediction should be more accurate than generate traffic from nothing. And for small cities which cannot provide a lot of historical data, there are many approaches applying meta-learning and transfer learning (e.g., metaST, STransGAN) that can potentially address this problem. Please explain the advantages of the proposed methods over these SOTA.
2. The paper did not explain many statements clearly, for example, it says one challenge is abnormal structure of roadnets, and it claims roadnet graphs in real-world do not always hold the basic law that ”neighbor nodes have similar features”, any example or reference?

**Questions:**

Please address the questions above.

---

### Official Review · Reviewer_M1Po · 2023-10-30

**Soundness:** 2 fair
**Presentation:** 2 fair
**Contribution:** 2 fair
**Rating:** 5
**Confidence:** 3

**Summary:**

TG-SCR (POI-based Traffic Generation model via Supervised Contrastive learning on Reconstructed graph) is a groundbreaking approach to traffic generation that overcomes the limitations of previous models. Unlike existing methods, TG-SCR combines both POI data and road network data to capture the intricate relationship between POIs and traffic more effectively. The model comprises two key modules: a graph reconstruction module and a POI supervised contrastive module. The graph reconstruction module reconstructs the original road network graph into a k-NN (k-Nearest Neighbors) graph. This transformation allows for a more comprehensive representation of the road network and enables a better understanding of traffic patterns. The POI supervised contrastive module enhances the representation of POI features by leveraging contrastive learning. This module ensures that the model can effectively capture the complex connections between POIs and traffic, resulting in more accurate traffic flow predictions. By integrating these modules, TG-SCR provides a more holistic and precise representation of traffic flows. The model's ability to capture the interplay between POIs and traffic facilitates better traffic generation and forecasting. TG-SCR represents a significant advancement in traffic modeling and paves the way for improved transportation planning, traffic management, and urban development.

**Strengths:**

1. Comprehensive integration of POI and road network data: One of the strengths of TG-SCR is its ability to effectively combine both POI data and road network data. By incorporating these two types of information, the model captures the complex relationship between POIs and traffic flows more accurately. This comprehensive integration allows for a more holistic understanding of traffic patterns and enhances the model's predictive capabilities.

2. Novel modules for improved modeling: TG-SCR introduces two innovative modules: the graph reconstruction module and the POI supervised contrastive module. The graph reconstruction module transforms the original road network graph into a k-NN graph, enabling a more detailed representation of the road network and better capturing its structural characteristics. The POI supervised contrastive module enhances the representation of POI features, considering their relationships with traffic. These novel modules address the limitations of previous models and contribute to a more precise and realistic modeling of traffic flows.

**Weaknesses:**

1. Dependency on quality and availability of input data: One potential weakness of TG-SCR is its reliance on the quality and availability of input data. The model requires accurate and up-to-date road network data as well as reliable and comprehensive POI data. If the input data is incomplete, inaccurate, or outdated, it may negatively impact the performance and reliability of the model. Additionally, if certain regions or areas have sparse or limited POI data, it may affect the model's ability to capture the full range of POI-traffic relationships accurately.

2. Complexity and computational requirements: Another potential weakness of TG-SCR is its complexity and computational requirements. The graph reconstruction module and the POI supervised contrastive module involve intricate operations and computations. Training and inference with such modules may require significant computational resources, including memory and processing power. This could limit the scalability and practical applicability of the model, particularly in scenarios where real-time or large-scale traffic generation is required. Additionally, the increased complexity of the model may also lead to longer training times, making it less feasible for time-sensitive applications.

**Questions:**

Questions are posted in weakness.

---

### Official Review · Reviewer_bmB5 · 2023-11-04

**Soundness:** 2 fair
**Presentation:** 2 fair
**Contribution:** 1 poor
**Rating:** 3
**Confidence:** 3

**Summary:**

This article addresses the problem of accurate trajectory prediction by incorporating contextual information into the prediction process. The previous methods have struggled to comprehend the impact of diverse and dynamic contextual information on trajectories. In contrast, this research proposes CATP, a Context-Aware Trajectory Prediction model. CATP consists of a manager model, multiple worker models, and a training mechanism inspired by competition symbiosis. Experimental evaluations on bird migration and video game datasets demonstrate that CATP outperforms four baselines supporting the effectiveness of the proposed method in trajectory prediction tasks.

**Strengths:**

This paper studies an interesting problem.

**Weaknesses:**

1. the novelty is limited. This paper seems to combine existing network together rather than proposing new methodologies. This does not meet the requirements of ICLR.
2. the technical contribution is limited. This paper simply combines several parts together, such as Graph Reconstruction Module, k-NN Graph Builder, POI Contrastive Module. The contribution is not enough.
3. the writing is not satisfactory.
4. some typos and format issues.

**Questions:**

n/a

---

### Official Review · Reviewer_ZfDL · 2023-11-06

**Soundness:** 2 fair
**Presentation:** 2 fair
**Contribution:** 3 good
**Rating:** 3
**Confidence:** 3

**Summary:**

This work constructs a DGNN for the purpose of traffic generation. Instead of using model based traffic generation simulation it utilizes real-world data alongside the DGNN approach. Instead of prediction, this work acts as NN traffic simulator. The approach is evaluated on 4 datasets with provision results.

**Strengths:**

- The introduced task looks is compared nicely with motion prediction task
- Results look s promising in terms of accuracy of the generation across 4 different datasets
- Figures are informative and increase the concept communication
- The overall work looks promising

**Weaknesses:**

- It would making reading better if POI term was introduced earlier in the paper.
- In section 3, 24 features vectors where mentioned. What is the exact the nature of these features?
- How exactly a "Traffic generation" task being evaluated? What is the target and how the other methods were considered in the exp. setup?
- Ablation study is limited, questions like the importance of the graph aggregator, choice of anchor node are missing.
- There is plenty of traffic generation methods that was not discussed in the work or compared to [1]. Model based approaches, SW to simulate traffic was not mentioned.
- Overall the work is provision in terms of motivation and results. There is a readability issues, some terms are not defines until later in the papers. The features (input/output) are not defined. The error and targets are not clear.
[1] Network Traffic Generation: A Survey and Methodology

**Questions:**

Regarding the law of general datasets "”the closer the points are, the closer the relationship is”, is not fully satisfied in real- world roadnet graph." Is there a line of work support such a claim?
- It is not always the case that connected nodes should have similar features, for example, in traffic vehicle and pedestrians can be connected with different sets of features.
- Is the loss function in 7 applied per all possible anchors?